# Shielded soft force sensors

Bekir Aksoy [1,2], Yufei Hao[1,2], Giulio Grasso[1], Krishna Manaswi Digumarti [1], Vito Cacucciolo [1] & Herbert Shea [1]

Force and strain sensors made of soft materials enable robots to interact intelligently with their surroundings. Capacitive sensing is widely adopted thanks to its low power consumption, fast response, and facile fabrication. Capacitive sensors are, however, susceptible to electromagnetic interference and proximity effects and thus require electrical shielding. Shielding has not been previously implemented in soft capacitive sensors due to the parasitic capacitance between the shield and sensing electrodes, which changes when the sensor is deformed. We address this crucial challenge by patterning the central sensing elastomer layer to control its compressibility. One design uses an ultrasoft silicone foam, and the other includes microchannels filled with liquid metal and air. The force resolution is sub-mN both in normal and shear directions, yet the sensor withstands large forces (>20 N), demonstrating a wide dynamic range. Performance is unaffected by nearby high DC and AC electric fields and even electric sparks.

Soft force and strain sensors provide the perception required for a broad range of fields, including human–robot interactions[1,2], wearable electronics[3,4], and health monitoring systems[5,6]. These soft sensors are mechanically compliant, can be mounted on nonplanar surfaces, and do not hinder motion. These features are key to enable soft machines and wearables to respond intelligently to their surroundings[7,8]. Amongst the possible sensing principles, including piezoelectric[9], optical[10,11], piezoresistive[12,13], and magnetic[14]; capacitive sensing gained prominence thanks to its low power consumption (driving current in μA range), relatively simple readout, fast response, and facile fabrication processes[15–18]. Capacitive sensors are, however, very susceptible to the motion or simply the presence of nearby objects, as well as to low-level electromagnetic (EM) interference. Despite numerous studies and the well-known effect of external interference, existing soft capacitive sensors lack the electromagnetic shielding needed to allow operation near moving conductors, motors, or other source of time-varying electric fields[19,20]. This restricts the use of soft capacitive sensors to laboratory settings with carefully controlled environments.

Capacitive force and strain sensors are of particular relevance for soft robotic systems[21–23], which often require independent measurement of pressure (contact) and proximity (non-contact) to safely interact with the environment[24]. Capacitive sensors without EM shielding cannot distinguish these two effects. EM shielding is widely used in electronic devices, and generally consists of metallic meshes or screens[25]. In the case of capacitive sensors, the shielding adds a parasitic capacitance between the shield and sensing electrodes. In rigid systems, this parasitic capacitance is constant, and can thus easily be compensated for in the readout, allowing stable and high signal to noise (SNR) operation. In contrast, for soft systems based on elastomers, the parasitic capacitance varies when a force is applied to the sensor because the distance between the shielding and the sensing electrode changes accordingly.

Several recent publications report shielded capacitive sensors based on a flexible printed circuit board (PCB) enclosed within a compliant conductive layer[26–28]. The external soft conductive layer to serve as both the shield and as one of the sensing electrodes. This configuration is intrinsically limited to measuring normal force. Measuring shear forces, requires sensing the relative change in capacitance of multiple (at least two) capacitors[29–31]. Even with additional electrodes, a design where the continuous shield also serves as one of the sensing electrodes cannot be adapted for shear force measurement. A solution to this is to decouple the shielding electrode from the sensing units, which is the approach we follow in this article. This introduces a new challenge in the form of parasitic capacitance between the shield and sensing electrodes, which is problematic because this capacitance changes when a force is applied to the sensor, as the sensors are completely soft. A key contribution of our work is two designs to

---

[1]Soft Transducers Laboratory (LMTS), Ecole Polytechnique Fédérale de Lausanne (EPFL), Neuchâtel 2000, Switzerland. [2]These authors contributed equally: Bekir Aksoy, Yufei Hao. ✉e-mail: herbert.shea@epfl.ch

overcome the effect of this parasitic capacitance and reliably measure both normal and shear forces in shielded soft sensors.

Shielding a soft sensor by simply coating it with a stretchable conductor leads to a system where the change in the parasitic capacitance can be larger than the change in the capacitance between the sensing electrodes. This effect can be reduced by tailoring the stiffnesses of the different layers in the sensor, e.g., using a softer material in the layer between sensing electrodes and a slightly stiffer material nearer the shielding. This presents several challenges, in particular when sensing not only normal force but also in-plane shear forces, as

sensitivity, softness, and mechanical robustness must be simultaneously achieved.

We present here two design approaches for shielded soft capacitive multi-axis force sensors, where we engineer the mechanical properties of the sensing layer by using an ultrasoft silicone foam and deformable microchannels filled with liquid metal and air (see Fig. 1). The sensors are unaffected by proximity effects, high electric fields, or even arcing in air between kV electrodes (see Fig. 1b, c). In these very demanding testing scenarios (see Supplementary Movie 1), the shielded sensors show negligible change in readout noise (standard deviation

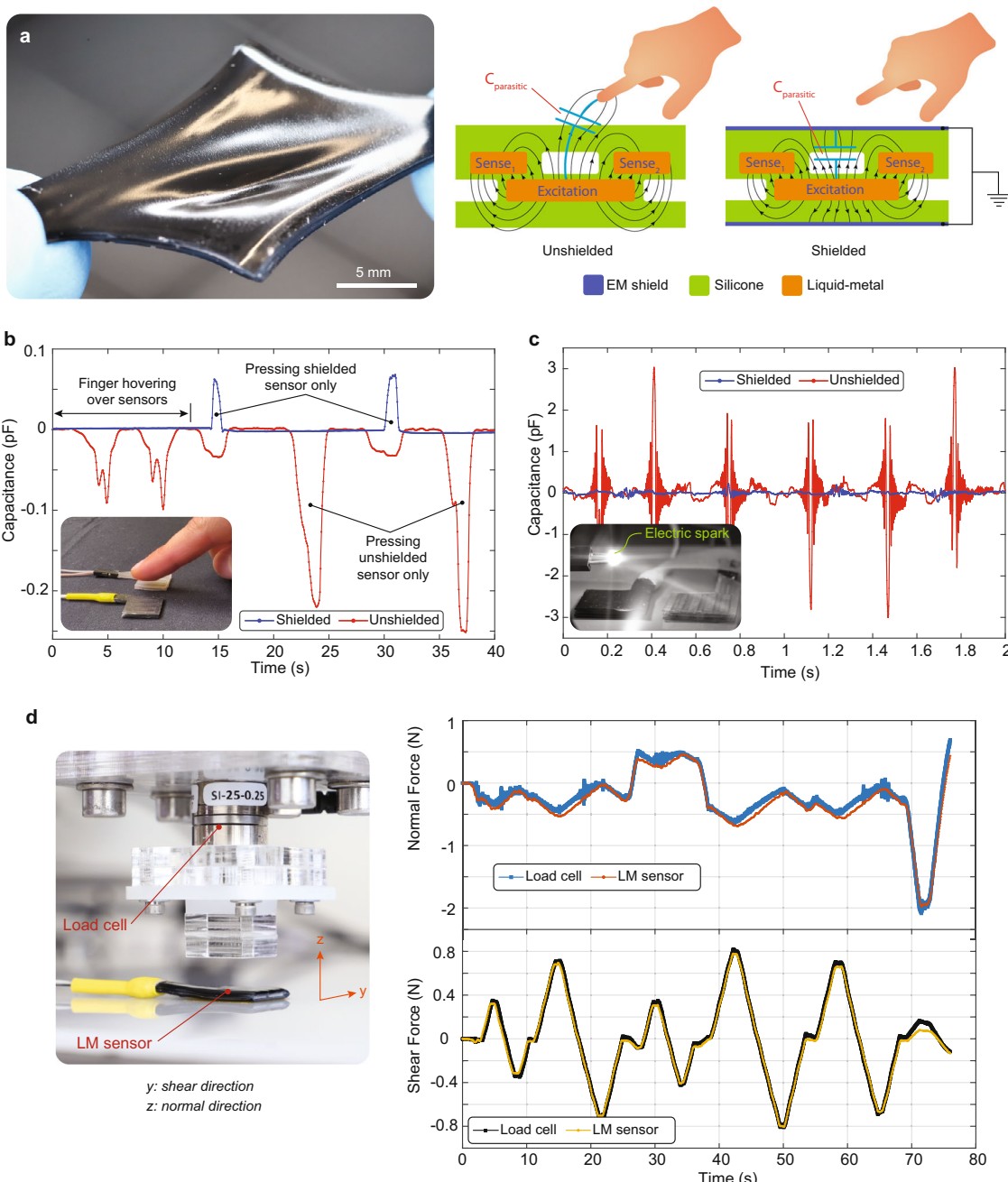

**Fig. 1 | Shielded soft sensors for accurate force measurement during soft interactions. a** Photograph of the sensor when stretched. Schematic cross-section, illustrating how the shielding make the sensor immune to proximity effects. **b** Comparison between the sensing capacitances of the shielded and unshielded sensor when a user's finger hovers over then sensors (not touching) and when the user presses on one of the sensors. The capacitance of the unshielded sensor changes due to this proximity effect even when the sensors is not touched. **c** Sparks juts over the sensors creates very high amplitude noise in the unshielded sensors, but very little in the shielded sensor. **d** Comparison of the force measured by a liquid metal soft sensor and a commercial multi-axis rigid load cell for a sequence of normal and shear forces. There is excellent agreement between the curves.

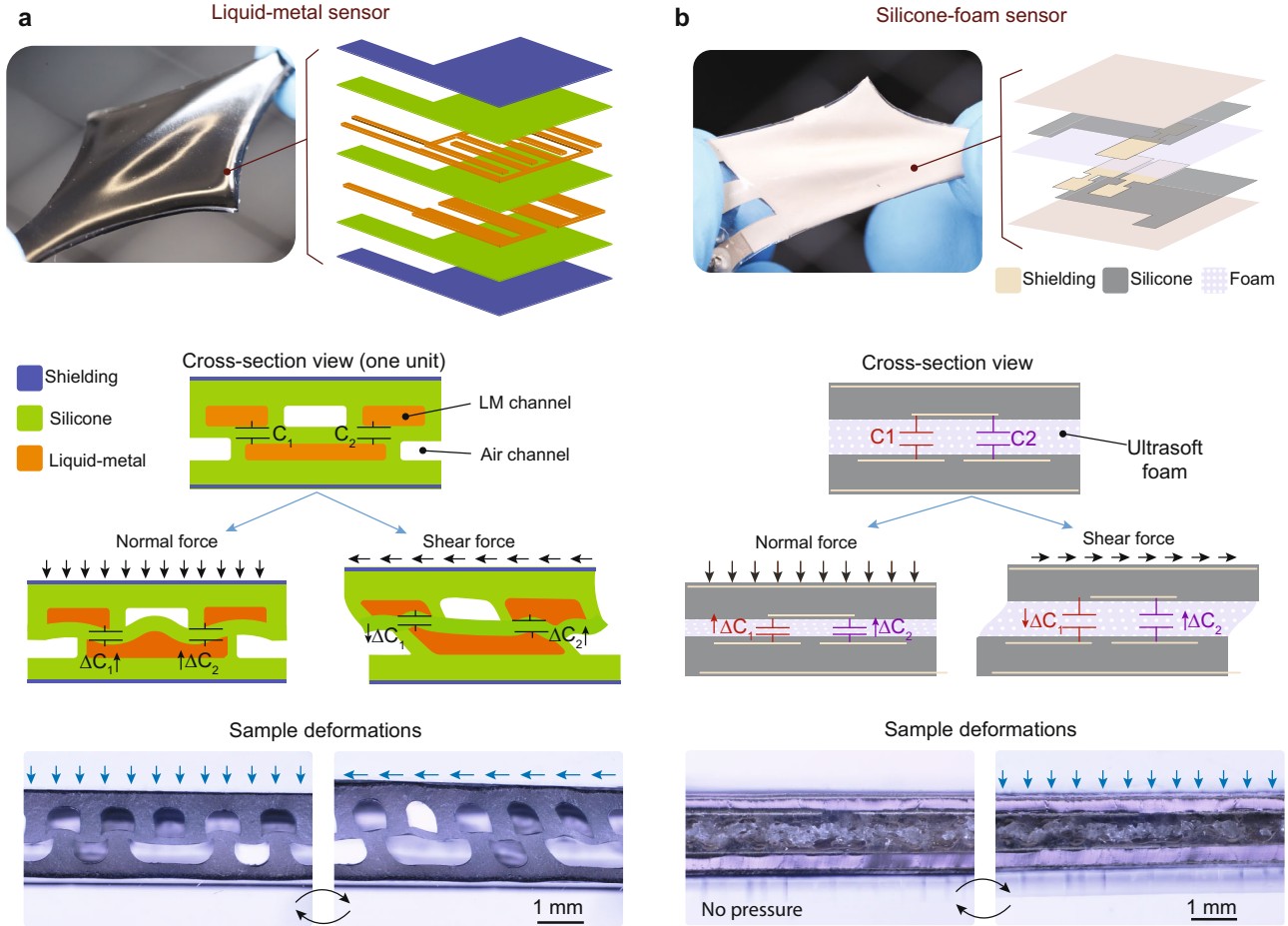

**Fig. 2 | Working principle of the shielded soft sensors. a** Exploded view of the liquid metal (LM) filled microchannel design. The multi-axis sensing is based on the relative capacitance change of two capacitors sharing a common excitation electrode. The cross-section views of a sensing unit demonstrate the change of these capacitances when loaded with normal or shear forces. The micrographs show the deformation of the micro-channels when pressed or sheared. **b** Exploded view of

the silicone-foam-based sensor. An ultrasoft foam is used as the dielectric material between the sensing electrodes. Similar to the LM design, the foam deforms easily under both shear and normal forces, allowing very sensitive force measurements. As seen from the micrographs, when a normal force is applied to the sensor surface, the foam undergoes a large deformation while the other components of the sensor remain mostly undeformed.

over 20 s of measurement), as opposed to unshielded sensors where the noise is order of magnitude larger than the signal, e.g., ± 0.10 fF versus ±23.39 fF. All materials used in making the sensors have Young's moduli less than 1 MPa. The total thickness is less than 2 mm. The sensor's low profile, flexibility, and stretchability enabled easy integration on nonplanar surfaces such as gloves or robotic hands.

Both sensor types have sub-mN sensitivity to the normal and shear components of applied loads. The comparison between our sensors and a multi-axis load cell (ATI Nano 17) shows a good overlapping for random loading scenarios (see Fig. 1d). In the foam-based sensors, a very soft silicone foam (SF) is used as a dielectric layer between the sensing electrodes, whereas the liquid-metal (LM) based sensors are made of silicone microfluidic channels filled with eutectic gallium-indium (EGaIn) alloy. Thanks to highly deformable sensing regions, the normal and shear forces can be simultaneously measured with sensitivities of 2.77 and 0.23 mNfF$^{-1}$, respectively. The noise of the shielded sensors at the rest position is ±0.09 fF, corresponding to ±0.25 mN of normal force and ±0.02 mN of shear force. Although the sensors measure small forces down to the mN range, they also sustain loads of tens of Newtons without degradation or failure. Thus, they provide a very large dynamic force range with a high signal-to-noise ratio.

We use the soft sensors to measure the interaction forces during the handling of various daily objects, from very soft water balloons to rigid steel rods and from electrically conductive aluminum plates to non-conductive acrylic cubes. The shielding successfully attenuates

any interference and enables accurate measurement of the grasping forces. The sensors provide accurate and clean output without being affected by either the presence of nearby high-voltage actuators (4.5 kVmm$^{-1}$) or by the electrical properties of the manipulated objects. The demonstrations with the sensors show their suitability for use in nearly any application scenario of soft machines, grippers and wearables.

## Results

### Design principles of capacitive-based multi-axis force sensing

We implement two different design strategies for our shielded soft sensors: a liquid-metal (LM) based microstructured architecture and an ultrasoft silicone-foam (SF) based design. Both sensors share a similar design paradigm: external compliant shielding electrodes enclosing a very soft sensing region and relatively stiffer passive sections elsewhere. This architecture is designed to sense very small forces while minimizing the effect of internal and external parasitic capacitances (see Supplementary Fig. 1). The relative positions of electrodes in the sensing region change when a force is applied. This leads to a change in capacitances. By design optimization, the change in capacitance per unit force is maximized while still ensuring a robust, stretchable, and manufacturable geometry.

The structural design of the sensors and their working mechanism are illustrated in Fig. 2. Three electrodes are used to sense normal and shear deformation. The LM design has a central section made of an elastomer (Ecoflex-0030) with two rows of microfluidic channels.

Some of the channels are filled with liquid-metal eutectic gallium–indium to serve as the electrodes. Liquid-metals is a promising class of materials for soft sensors[32,33] and actuators[34,35] because they can undergo large deformations while maintaining their high electrical conductivity. The remaining channels are intentionally left empty (i.e., with air) to provide space for the displacement of the liquid-metal when the sensor is pressed (see Supplementary Note 1 and Supplementary Fig. 2). These empty channels compensate for the incompressibility of the liquid-metal and of the elastomer.

To choose an optimum electrode layout that is sensitive to both normal and shear forces, candidate designs are evaluated in COMSOL (Supplementary Fig. 3 and Supplementary Note 2). Three parameters are considered: the initial capacitances in the undeformed state, the change in capacitances per unit change in normal force, and the change in capacitances per unit change in shear force. The initial capacitances need to be in a suitable range (< 17 pF) for it to be measured by the portable and low-cost readout circuit we use. Designs with higher changes in capacitance when a force is applied have higher sensitivity and are therefore preferred. The layout with the best performance across three parameters is one in which the top row of electrodes is horizontally shifted with respect to the bottom row by half the channel width (see Fig. 2a). In this configuration, there is a misalignment of the vertical walls of the channels, which reduces the effective mechanical stiffness of the channels and therefore amplifies the deformation under an external load (easier to compress) which enhances the sensing performance. Another feature of this design is that the sensors are able to withstand large normal forces without being damaged. This is because the channels with liquid metal can take up the space of the air pockets under these loads and do not allow further compression. When the load is removed, the channels return to their original configurations.

A cross-section view of a LM sensing unit is shown in Fig. 2a. Each unit measures 3.2 mm in width, 2 mm in height and spans a depth of 14.6 mm. There is one electrode in the bottom row and two electrodes in the top row. The bottom electrode is used for excitation and the top electrodes are used for sensing the capacitance. This configuration defines two capacitances ($C_1$ and $C_2$), which enables us to simultaneously sense the shear and normal components of the applied forces based on the relative change of their capacitances. Other configurations of excitation and sensing electrodes are less sensitive than the one shown here and do not allow the simultaneous measurement of shear or normal force. The LM sensors have four of these sensing units which are connected in parallel, i.e., the equivalent capacitance of the sensor is equal to the sum of these sensing units. Because these units cover the entire sensor area, the local forces are picked up by one or multiple units depending on the surface area of the load. For simplicity of presentation, we only discuss the design of a single sensing unit. The deformation of the channels under normal and shear forces is illustrated in Fig. 2a. When a normal force is applied, both capacitances increase. Under pure shear force, one of the capacitances increases while the other one decreases. The micrographs depict the deformation of the channels when loaded in the normal and shear directions. The micrographs are captured before filling the channels with liquid-metal.

Similar to the LM sensor, the design principle of the silicone-foam sensor is based on having functional layers with different stiffnesses, and three compliant excitation and sensing electrodes. It is designed to have a deformable material between the electrodes and relatively stiffer structure elsewhere (see Fig. 2b). This is accomplished by using a 600 μm thick ultrasoft foam. The elastic modulus of the porous silicone foam is about 7 kPa, whereas the bulk silicone layer has a modulus of 1 MPa (see Supplementary Fig. 4a). The foam layer is sandwiched between two composite layers of electrodes + silicone + shielding. Figure 2b shows the oblique and cross-sectional views of these layers. The electrodes and the shielding are made of silver-filled composition (Ag 520 EI from Chimet S.p.A.), and the silicone layer is made of polydimethylsiloxane (Sylgard 186 from Dow Inc.).

The sensitivity of both sensors depends on the deformability of the sensing region. In the SF sensor, this is optimized by tailoring the foam stiffness, whereas in the LM sensor it is achieved through structural design. Supplementary Fig. 4a depicts the deformation of sensing and passive region for SF sensor. The foam is 143 times softer than the other layers, and therefore most of the deformation is confined to the foam when pressing the sensor. Although making softer foam leads to higher sensitivity, it makes them weak against external forces. In LM sensors, we use relative deformation as a proxy for deformability of the sensing and passive regions. The relative deformation of these regions in the LM sensor is plotted as functions of the applied normal force. For small forces (<100 mN), the ratio between these regions is 30 and decreases down to 20 at 450 mN of force, showing the stiffening of the sensing region (see Supplementary Fig. 4b). Thanks to this stiffening, the LM sensors can survive at very high forces.

## Design considerations of shielding

Capacitive force sensors are affected by objects interacting with the fringing field between the electrodes. To ensure that the sensors only measure the applied external force (i.e., the deformation of the electrodes due to this force) and not a change in apparent capacitance due to electrical interference, the sensing region is shielded using conductive silicones (see Supplementary Movie 1). Shielding is primarily required on the exposed surfaces of the passive regions. As seen in the cross-sectional view of Fig. 2, the electrodes of the sensing region are close to the outer surface (<270 μm), and the fringing fields extending out of the surface are most likely to be influenced by external objects (see Supplementary Fig. 1). On the sides of the sensor, the electrodes are farther away from the outer surface (>1.7 mm) and there is no effect on the fringing field from external objects.

The properties of the shielding material need to be chosen with some care. It is desirable to use a material that has an elasticity modulus comparable to the material of the sensors. This is necessary to minimally affect the stiffness of the entire structure. Our sensors are shielded using carbon-loaded silicone (CB) in the LM sensors and silver-filled (Ag) composition in the SF sensors. They are coated in thin layers (<54 μm) and have lower moduli of elasticity (<1 MPa) than the structural materials. They, therefore, only negligibly increase the stiffness of the structure.

The shielding layers lead to parasitic capacitances ($C_{parasitic}$) between the sensing electrode and the shield (see Fig. 1a). Since the sensors are made of soft materials, the passive region between the sensing electrode and shield also deforms and this parasitic capacitance changes when the sensor is pressed. This is problematic as a change in the parasitic capacitance affects the measurement of the capacitance between the sensing and excitation electrodes. Simulations in Supplementary Fig. 5 compare the sensing and parasitic capacitances of two sensors where their passive regions have different stiffnesses (also see Supplementary Note 3). The simulations show that a softer passive region results in a parasitic capacitance that changes by a large amount, e.g., 0.25 fFmN$^{-1}$ versus 0.04 fFmN$^{-1}$. It is desirable to minimize the change in the parasitic field under mechanical load. In our sensor designs, the change in the sensing capacitances is 11 times higher than the parasitic ones. Therefore, the change in the parasitic capacitances is negligible.

## Experimental evaluation of shielding on the performance of the sensor

The shielding performance is quantified for two scenarios: motion of a metal object a few mm away from the sensor, and 1.5 kV applied to interdigitated electrodes 120 μm above the sensor. The effectiveness of the shielding layer for blocking proximity effects is evaluated by moving a metal plate towards the sensor, starting at a distance of

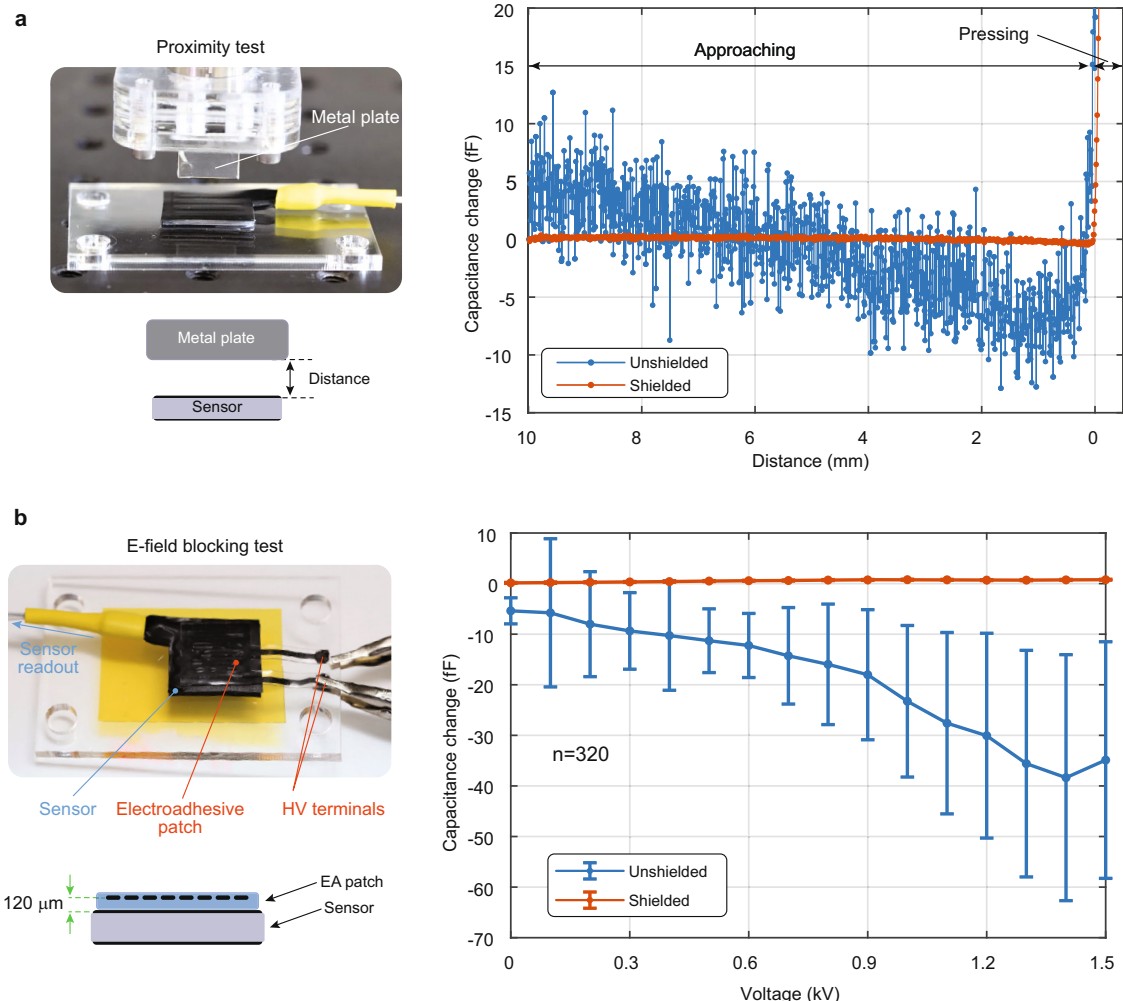

**Fig. 3 | Noise comparison between the shielded and unshielded sensor. a** The capacitance change of a shielded and an unshielded sensors when approaching a metal plate. **b** The capacitance changes due to the fringing electric field of a nearby electroadhesive patch that is operated at AC voltage up to 1.5 kV. In both scenarios the shielded sensors are not affected by these interferences whereas the unshielded sensors are. Error bars represent the mean ± 1 standard deviation over 20 s of measurement ($n$ = 320 for each voltage step).

10 mm and moving until it contacts (see Fig. 3a). The metal plate is attached to a load cell and mounted on a motorized stage. Contact between the sensor and the metal plate is detected by this load cell. After touching the sensor surface, the plate is pushed a bit further to show the clear capacitance increase due to compression. Figure 3a compares the readouts of shielded and unshielded sensors. The unshielded sensor has noisier data and shows a large significant capacitance change when approaching the sensor surface, i.e., a change in capacitance with no applied force. The benefit of the shielding layer is evident in terms of the sensor noise, which drops from 5.4 ± 2.6 fF in the unshielded design to 0.14 ± 0.09 fF in the shielded design.

In the second scenario, the shielding performance is tested by placing a high-voltage electroadhesive (EA) patch of interdigitated electrodes directly on the top of the sensor (see Fig. 3b). The patch is actuated with a 20 Hz AC voltage, ramped up to 1.5 kV. The sensing capacitances are measured every 100 V for 20 s at a sampling rate of 16 Hz. Figure 3b compares the capacitance change of the shielded and unshielded sensors. The error bars correspond to one standard deviation from the mean. The electrostatic actuation of the patch significantly changes the readout of the unshielded sensors, leading to high measurement errors. The shielded sensor effectively blocks the fringing electric field of the patch. The tests are carried out for both

sensor types. Figure 3 presents the results of the LM sensors whereas the data of SF sensors is shown in Supplementary Fig. 6. Both sensors have similar shielding effectiveness.

The shielding effectiveness of the carbon-loaded silicone, silver-based ink, and bare silicone layers is measured using a network analyzer (E5071C from Agilent Technologies). Two ports of the network analyzer are fixed 0.5 mm apart, as shown in Supplementary Fig. 7a. The membranes are placed between these probes, and the $S_{21}$ (i.e., transmission) characteristics are measured as the frequency is swept from 10 to 200 MHz. A baseline test is done without any membrane between the probes. Then the bare, carbon-based, and silver-based silicone layers are used. The silver-based shielding performs the best by attenuating the signal by 19 to 30 dB, whereas carbon-based shielding reduces the signal by 18 to 20 dB.

**Sensor force readout calibration**

To determine the relation between the applied force and change of the two capacitances, we calibrate the sensors using known forces. The mechanical deformations due to the normal and shear components of the force are not independent. The change in capacitance for a given shear deformation depends on the magnitude of vertical deformation (see Supplementary Note 4 for capacitive coupling between normal and shear deformation). Calibrating the sensor allows us to decouple

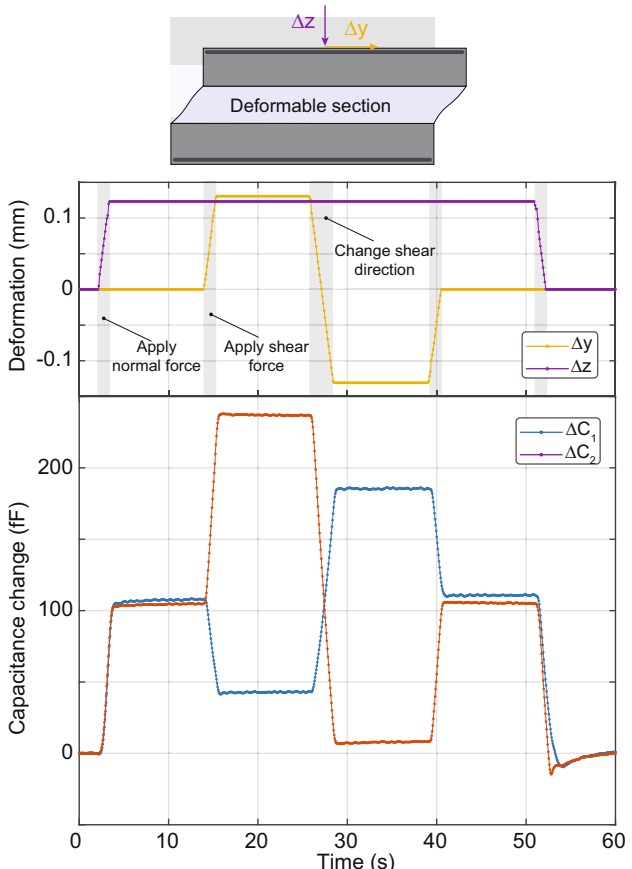

**Fig. 4 | Capacitance change of a liquid metal sensor under simple normal and shear deformations.** The schematic shows the shear (*y*) and normal (*z*) force directions. The top graph plots the time dependence of the normal and shear deformations applied to the sensor. The sensor is first compressed by applying deformation in the −*z* direction (normal force). It is then additionally deformed in the +*y* direction (+ normal + shear). The sensor is then deformed in the −*y* direction while the normal force is kept constant (+normal − shear). The shear force and then the normal force are removed. The time evolution of the capacitances is plotted in the bottom graph as the sensor goes through these steps. When the normal force is applied, both capacitances increase. The shear force, however, changes the capacitances in the opposite ways, e.g., decreasing $\Delta C_1$ while increasing $\Delta C_2$. Similarly, flipping the shear force direction increases $\Delta C_1$ and decreases $\Delta C_2$.

the measurements and thus determine each force component. We apply a sequence of increasing normal and shear forces to get a map of capacitance change vs applied forces.

A typical sensor readout for a simplified cycle of applying normal and shear forces is shown in Fig. 4. When the normal force is applied, it vertically compresses the sensor, increasing both capacitances $C_1$ and $C_2$. As seen from the graph, 120 μm vertical displacement (Δ*z*), corresponding to 1.28 N of force, causes approximately +105 fF change in both capacitances (see Supplementary Fig. 7 for force values). When a shear deformation is then applied in the horizontal direction (Δ*y*), one of the capacitances increases, while the other one decreases, e.g., applying 0.65 N of shear force, increases $C_2$ by 132 fF and decreases $C_1$ by 62 fF. Applying the shear force in the opposite direction decreases $C_2$ and increases $C_1$. Figure 4 shows how the capacitances change when the shear is applied in the opposite directions. Once the forces are removed, the initial state is recovered, and the capacitances return to their initial values.

A motorized stage with an attached multi-axis load cell (ATI Nano 17) is used to apply the sequence of displacement needed for the calibration process. The forces are applied via a probe with a surface area of 12 mm × 12 mm. The calibration steps are illustrated in Fig. 5a.

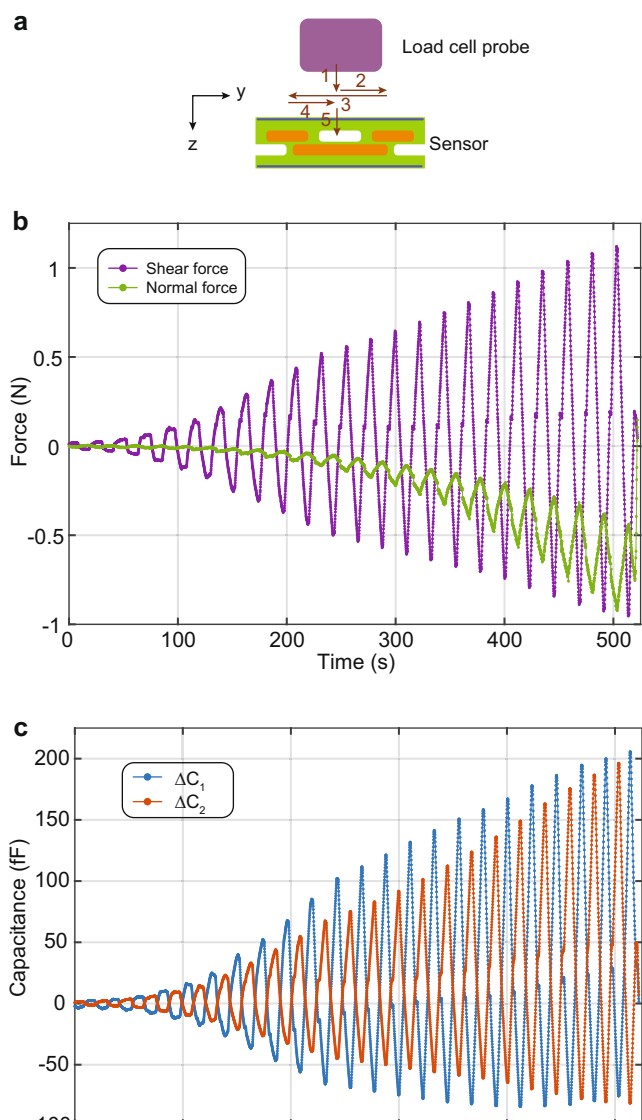

**Fig. 5 | Sensor characterization procedure. a** Sequence of motion that is repeated as the stage is moved in the *z* direction. **b** The normal and shear forces measured (load cell) during the characterization vs. time. **c** The time evolution of the two capacitances measured by the liquid metal sensor.

We first apply 5 μm displacement in the normal direction (step 1). Then a shear displacement of ± 0.2 mm is applied in the horizontal directions (steps 2 and 3). For each additional vertical step of 5 μm, we apply the shear in the horizontal directions with the same displacement of ±0.2 mm. We repeat these steps until a total vertical displacement of 0.2 mm is reached. This way, we cover the forces and the corresponding capacitance values for ±2 N. The capacitances are measured using a capacitance meter (from JLM innovation). The analog output of the force sensors and capacitance meter are recorded using a DAQ card (NI USB-6210 from National Instruments) and a custom Python script.

Figure 5b, c shows the force and capacitance data measured during the sensor characterization, representing how the applied force and measured capacitances change at each step. For our sensor design, the normal force increases with the vertical displacement, and the shear force changes according to the horizontal deformation. However, these two force components are not completely decoupled (see Supplementary Note 4 and Supplementary Fig. 8 for more details).

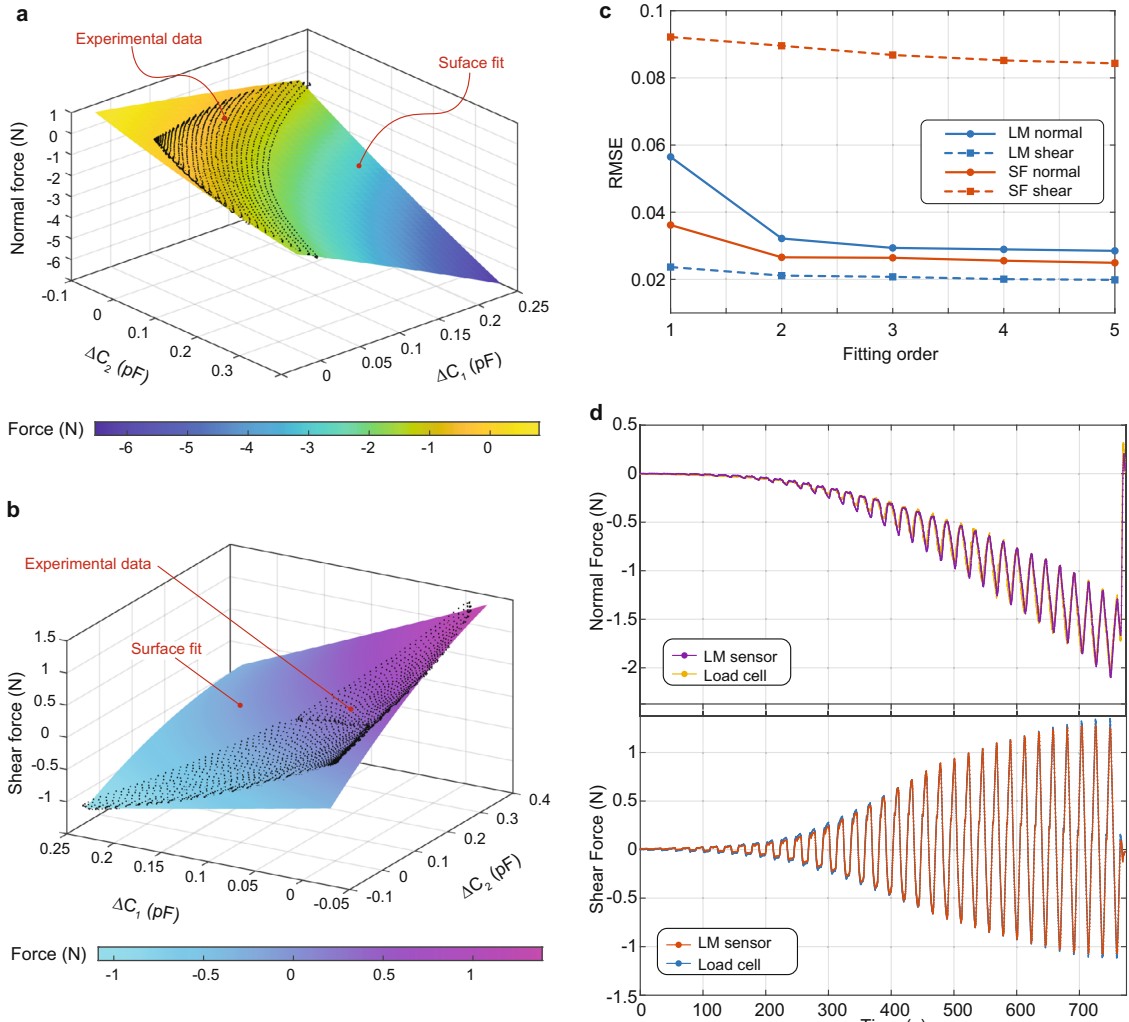

**Fig. 6 | Force-capacitance conversion and comparison between the liquid metal sensor and the load cell. a** The measured normal force is plotted as a function of measured capacitance changes, $\Delta C_1$ and $\Delta C_2$. A surface is fitted to this data. The 3D plot shows the experimental data (black dots) and the surface fit with second order ($n = 2$). The surface fit allows us to extract the coefficients for the capacitance–force conversion, and, therefore to formulate the normal force as a function of $\Delta C_1$ and

$\Delta C_2$. **b** The same fitting method is followed for the shear component where in this case the shear force is plotted as a function of the capacitance change. **c** The root mean square error (RMSE in N) of different sensor types are plotted vs. the fitting order. Based on these results, we choose $n = 2$ as the fitting order. **d** The shear and normal forces from the load-cell and from the soft sensor showing excellent agreement for both force components.

For instance, the same horizontal deformation generates different shear forces depending on the vertical deformation. At $t = 100$ s (vertical deformation is 25 µm) the shear force changes between ±0.15 N for the horizontal displacement of ±0.2 mm, and the same horizontal deformation causes ±1.20 N of shear force at $t = 500$ s (vertical deformation is 120 µm). In other words, the shear force for the same horizontal deformation depends on the magnitude of the normal force. This can be clearly seen in the force plots shown in Fig. 5b. The similar pattern is observed in the capacitance change as well. For the same shear displacement, the capacitances have small changes when the vertical deformation is small and large changes when the vertical deformation is high.

To extract the normal and shear components of the applied force as functions of the capacitance changes, $\Delta C_1$ and $\Delta C_2$, we use a surface fitting method as summarized in Fig. 6. First, we plot the measured normal force as functions of measured $\Delta C_1$ and $\Delta C_2$. We then fit a surface to this experimental data to extract an expression for the normal force as a function of capacitance change, i.e., $F_{normal}$ ($\Delta C_1$, $\Delta C_2$). Figure 6a shows a surface fit on a 3D plot of the normal force. The black dots are the experimentally obtained normal force–capacitance

data. We utilize surface fits with different orders, from 1 to 5. For each fitting order, we calculate root mean squared error (RMSE in N) between the experimental data and surface fit. Then we repeat the same fitting procedures for the shear force. A sample surface fit of the shear force is shown in Fig. 6b. Using the formula of these surface fits, we can express the normal and shear forces as functions of the capacitances change, e.g., $F_{shear}$ ($\Delta C_1$, $\Delta C_2$) and $F_{normal}$ ($\Delta C_1$, $\Delta C_2$). A sample 2nd order fitting of the normal force would have a formula as

$$F_{normal} = a\triangle C_1 + b\triangle C_2 + c\triangle C_1^2 + d\triangle C_1\triangle C_2 + e\triangle C_2^2 \qquad (1)$$

where the force is in Newton and the capacitance change is in pF. The conversion coefficients $a, b, c, d,$ and $e$ correspond to the coefficients of the surface fit. In most cases, increasing the fitting order above 2 did not further decrease the RMSE (see Fig. 6c). We thus choose the 2nd order fitting for our sensors. For the LM sensor, for instance, the RMSE in fit is around 0.03 N in the normal direction and 0.02 N in the shear direction. Using the conversion coefficients of the 2nd order fitting we replot the normal and shear forces and compared them with the load cell data. Figure 6d compares the force measurement between our soft

sensor and the commercial load cell. Both for normal and shear forces, we have an excellent agreement between the calibrated LM sensor and commercial load cell. Since both surface plots of shear and normal forces are monotonic, we can decouple these force components and accurately formulate each one using the surface fitting method.

The elastic hysteresis of the sensors is measured using a uniaxial test machine (see Supplementary Fig. 10a). The sensor response is tested at two different speeds: 0.002 and 0.02 mms$^{-1}$. The sensor shows a small degree of elastic hysteresis of 5% (the maximum of the difference in the ordinate expressed as a percentage over the range of the ordinate) between the loading and unloading due to viscoelasticity of the PDMS (see Supplementary Fig. 10b). The hysteresis behavior is observed to be identical for both tested speeds. In the case of capacitance change as a function of displacement, a 2% hysteresis is observed (see Supplementary Fig. 10c). Once again, there is no rate dependence. In all cases, the force and capacitance return to initial values when the displacement is recovered, showing no permanent deformation. The sensor shows low hysteresis and no permanent set, important features for soft sensors.

The sensors are resilient to cyclic loading. The capacitance of the sensors is measured over thousands of cycles over 30 h for an applied force of 2 N on the LM sensor and of 0.8 N on the SF sensor (see Supplementary Fig. 11). The maximum deviation in the capacitance from initial value after $10^4$ cycles is <10 fF in the LM sensor and <12 fF in the SF sensor. These variations correspond to 35 mN in the LM sensor and 53 mN in the SF sensor. The sensitivity of the sensors to bending is measured using an experimental setup where the bending radius of curvature of the sensor could be continuously varied. The experimental setup and plot of capacitance as a function of the bending curvature are shown in Supplementary Fig. 12. The test is repeated for two configurations: the bending axis is orthogonal to the shear direction, and the bending axis is parallel to the shear direction. The measured change of capacitances per curvature change is 2.73 and 1.73 pFmm for orthogonal and parallel configurations, respectively. For a 100 mm change in the radius of curvature, the bending causes a force error of approximately 85 mN in orthogonal configuration and 46 mN in parallel configuration.

If the sensor is placed on a surface of fixed curvature, the sensor can simply be calibrated for this given curvature, giving the same accuracy in normal and shear force components as for the flat state. If we, however, use the sensor in a scenario where the sensor is being simultaneously bent and pressed at the same time, the capacitance changes due to bending deformation then need to be taken into account for accurate measurements. Different approaches can be implemented to compensate this effect, such as attaching an additional sensor on the back of the beam that can sense the curvature only. This way, both the applied load and the curvature can be measured simultaneously.

For a given device area, the LM and SF sensors have comparable performance in terms of force resolution, softness, and resilience under cyclic loading. Both sensor types can withstand high normal forces (>20 N). However, under high shear forces, the silicone foam sensors fail at approximately 1.5 N, while the liquid-metal sensors can sustain shear loads of over 14 N. The LM structure can more readily be tailored to provide both the required low stiffness and also good robustness.

Fabrication complexity is similar for both sensors, with molding, casting, and bonding steps. The LM sensors require slightly more careful handling during the bonding and the liquid-metal filling steps because the channels must be perfectly sealed. The sensor fabrication process is reproducible: there is less than 2% difference in capacitance between devices from one batch of LM devices. The total thickness of the LM sensor is around 2.2 mm whereas the SF sensors have a thickness of around 1.8 mm.

The LM sensors have better durability than the SF sensors under both normal and shear loads. We, therefore, use the LM sensors in our demonstrations.

## Force measurements during interaction with soft objects

This section covers the demonstrations of the sensor in different use cases: as a sensor for robotic grippers, as a sensory skin, and as a sensor to measure the interaction forces between a paper strip and an electroadhesive surface. To demonstrate that the sensor can be used to measure interaction forces when handling soft objects, we mount it on one finger of a two-fingered robotic gripper and manipulated a water balloon (see Fig. 7a and Supplementary Movie 2). The fingers first grasp the water balloon, lift it, rotate it by 135°, rotated back to initial orientation, and finally release it. Frames from Supplementary Movie 2 show the different states of the pick-and-place action. The force components measured by the sensor are plotted vs. time. When the object is first grasped, both the normal and shear force increase. The normal force increases due to compression of the balloon between the fingers. When the balloon is squeezed, the water pushes the sensor in an upward direction (it cannot go downward because of the solid base). This causes a change in shear force when grasping the object. Once the object is lifted, the normal force slightly changes, whereas the shear force jumps up due to the gravitational force of the object acting on the sensor. The object is then rotated clockwise with an angle of 135°. During the rotation, both the normal and shear forces change as expected due to the change of the gravitational force component on the sensor. Since the finger with the attached sensor stays on the top during this rotation, both force components decrease. When rotating back to the initial configuration, the initial force values are recovered. The sensor accurately measures the interaction forces and senses the change in the forces when the griper rotates.

The sensors can be integrated on the human hand as a sensory skin. For this demonstration, the sensor is mounted on a user's thumb while picking and placing different objects (see Fig. 7b). The normal and shear forces change according to the type and the weight of the grasped object. For deformable water balloons, for example, the normal force is low compared to when picking up rigid objects. A higher shear force is measured when heavier objects are lifted, e.g., 130 g water balloon has 0.7 N of shear force, while for 106 g water balloon this is around 0.5 N. Supplementary Movie 3 shows the force measurement during various daily life tasks such as screwing bolts, peeling fruits, cutting with knife, typing, and cleaning. The demonstrations of different use cases show reliable sensor output independent of the testing conditions.

In the final demonstration, we attach an electroadhesive patch to the sensor and operate it at AC voltages from 0 to 1 kV (see Supplementary Movie 4). The goal of this demonstration is to show that the sensor can accurately detect the adhesion forces between the electroadhesive surface and an object without being disturbed by the high voltage electric field (3 kVmm$^{-1}$) in close proximity to the sensor (120 μm). A paper strip is placed directly on the EA skin. One end of the strip is attached to a commercial load cell which is mounted on a motorized stage (see Fig. 8). The paper is pulled and pushed using this stage, making the paper slide on the patch. The load cell provides a reference against which to compare our soft sensor. When the electroadhesive patch is off, there is a small shear force between the EA patch surface and the paper due to the mechanical friction caused by the weight of the paper. When the high voltage is turned on, the paper starts adhering to the patch due to the induced surface charges on the paper caused by the high electric field[36]. Once the high voltage is on, the shear force increases, causing high traction between the patch and the paper strip.

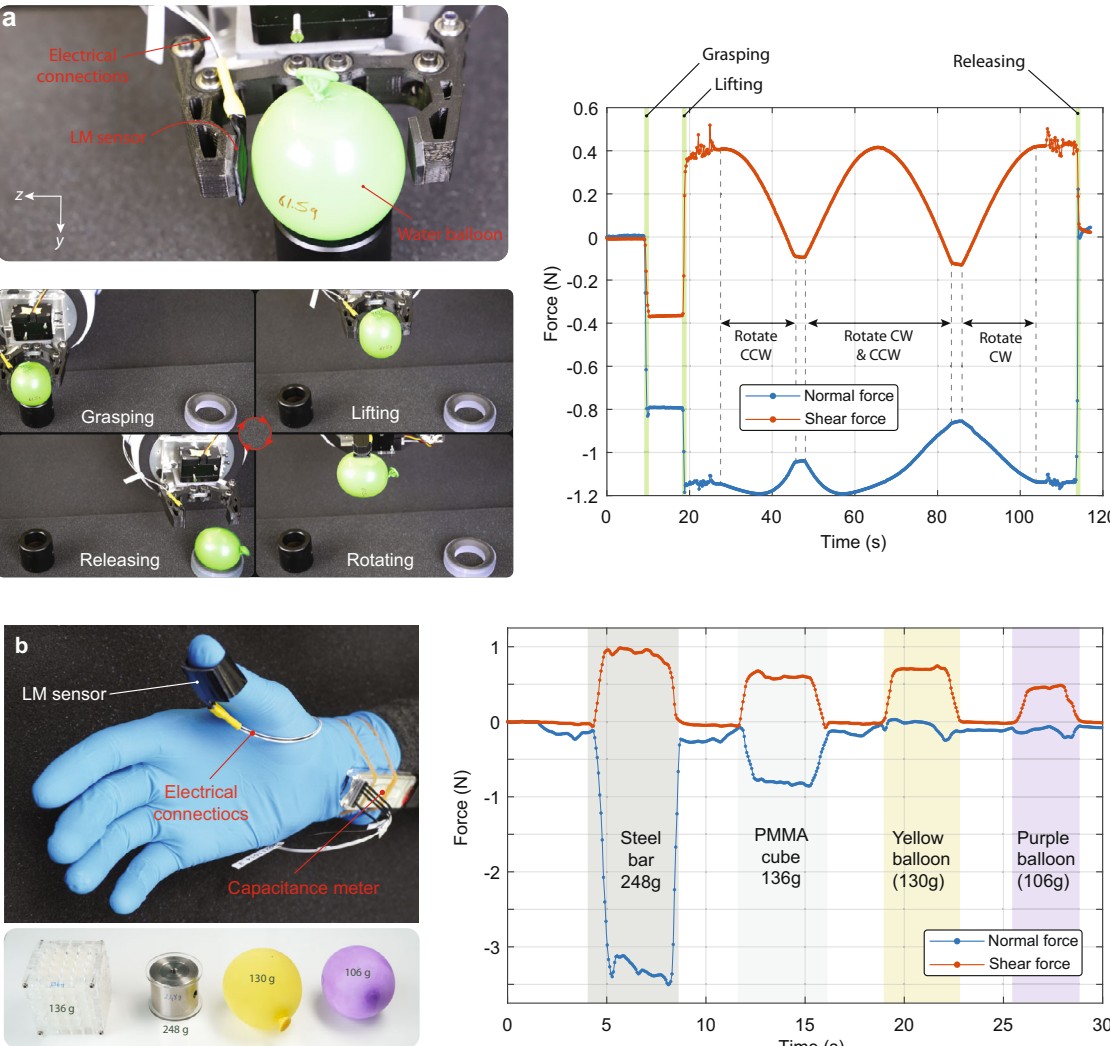

**Fig. 7 | interaction forces measured by the soft sensor during the manipulation of various objects. a** The soft sensor is attached to a finger of a robotic gripper and is used to measure the normal and shear forces when a water balloon is grasped, lifted, rotated, and released. The plot shows the time evolution of the force components during this manipulation. **b** The soft sensor mounted on a thumb, measuring the interaction forces when picking up and handling different objects. A conductive steel bar (248 g), an acrylic cube (136 g), and two water balloons (130 g and 106 g) are used in this demonstration. The softer objects have lower normal force whiles the rigid object has higher values in compression.

The traction force between the paper strip and the EA patch is simultaneously measured by the load cell and soft sensor. As seen in Fig. 8, both sensors measure nearly identical shear forces for all phases of the test, i.e., pulling, pushing, EA on and EA off. For example, when the sliding direction of the paper strip is reversed at $t = 75$ s (no voltage), the shear force value has a small jump from negative to positive value and these small amplitudes are clearly picked up by the soft sensor. When the paper is pushed, and the voltage is on, the paper buckles (see Supplementary Movie 4). When the paper buckles, it partially detaches from the skin surface and causes less adhesion. This action is observed in the pushing case where the traction force has sudden drops.

## Discussion

We report the design, fabrication, calibration and validation of shielded capacitive soft force sensors where the shielding does not degrade sensing by ensuring the parasitic capacitance of the shield remains constant when the sensor is deformed. Our devices have high sensitivity both in the normal (2.77 mNfF$^{-1}$) as well as one shear direction (0.23 mNfF$^{-1}$ numbers) and with very low noise <0.09 fF. The sensors are less than 2 mm thick, and are made of stretchable

soft materials (Young's modulus of less than 1 MPa), allowing for shape adaptation to many objects. Our sensors have a marked advantage over existing designs of capacitive sensors in that they are shield, making them insensitive to motion of external objects and to external electric fields. This allows the sensors to be used in numerous applications, such as mounted on robotic grippers or on hands, where state of the art capacitive sensors are unusable due to excessive noise.

We demonstrate our sensors in multiple scenarios, including as sensory skins to quantify the grasping forces of common objects, as traction sensors to measure the adhesion force of high-voltage electroadhesive patches, and to measure the interacting forces for the robotic grippers. Even in extreme operating conditions in close proximity (<0.12 mm) to high voltage fields (4.5 kVmm$^{-1}$) or electric sparks, the sensor operates well.

Our soft capacitive sensors are a composite structure with layers of different stiffness, enabling measuring forces down to the sub-mN range. The sensors can sustain forces greater than 20 N without deterioration. The load capacity of the sensors can be further increased by using stiffer materials or by changing the dimensions of the layers. For example, in the LM sensors, most of the deformation is

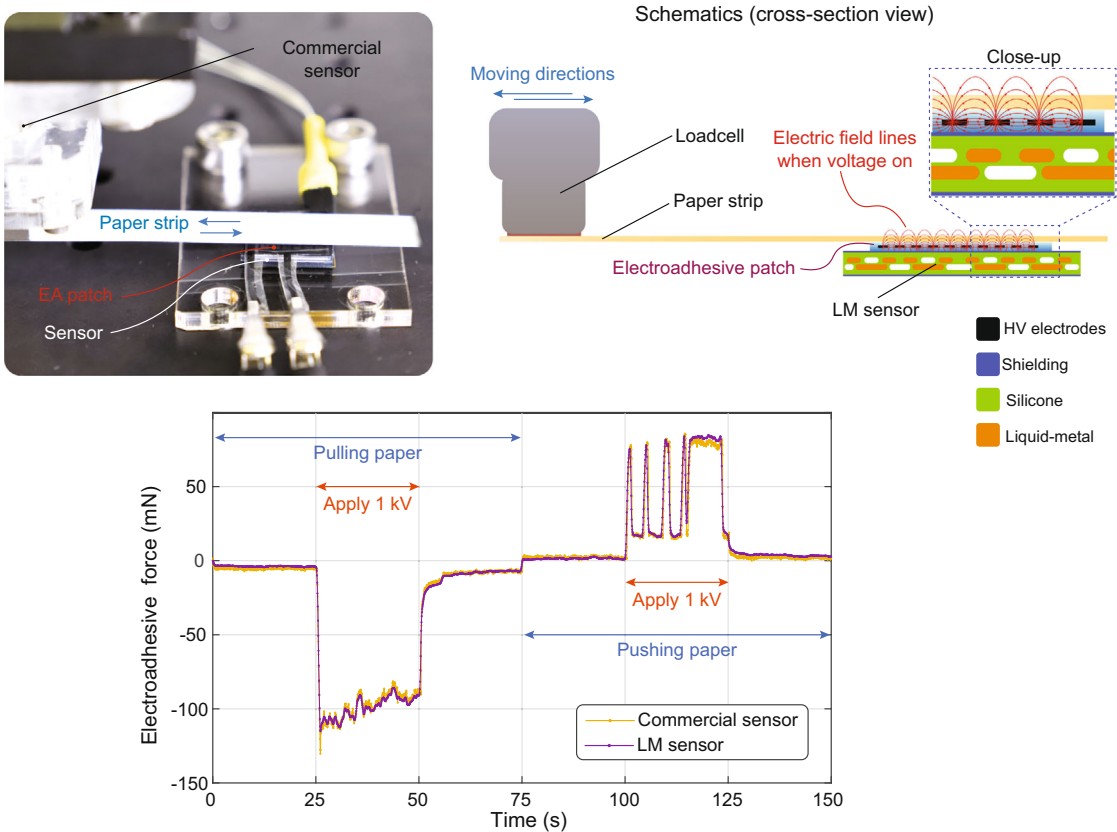

**Fig. 8 | The soft sensors accurately measure the traction force between the paper strip and electroadhesive surface.** The photographs show the experimental setup and the close-up of the electroadhesive patch (EA), which operated at 1 kV AC during pulling and pushing tests. The EA patch is directly placed on top of the sensor. The voltage is turned on between 25 and 50 s (pulling) and between 100 and 125 s (pushing). The graph compares the traction forces measured by the commercial load cell and the LM sensor. Although the EA patch is placed at a very closed location and operated at high voltage (3 kVmm⁻¹ of electric field), the shielding effectively blocked the fringing electric field of this patch and accurately measured the traction force.

confined to the middle layer and increasing the thickness of this layer will increase the load capacity, at the cost of reduced sensitivity. The LM sensor design allows easy modifications to tune the load capacity and sensitivity based on application requirements.

# Methods

## Foam fabrication

To fabricate the ultrasoft elastomer foam, sugar is crushed and winnowed with two mesh filters to ensure a particles size between 125 and 190 $\mu$m. The sugar is mixed with DI water at a weight ratio of 25:1. The resulting paste is pressed into a cubic mold with a depth of 0.5 mm. After removing any excess sugar paste, the mold is put in the oven at 80 °C for 2 h to evaporate the water, forming a sugar block. Uncured Ecoflex 0030 (ratio of part A and part B is 1:1) is poured on the sugar block and put under vacuum several times to ensure the elastomer infiltrates the sugar block. Any excess silicone rubber is squeezed out of the mold by pressing on the mixture with a heavy cover. After 4 h to cure the Ecoflex 0030 at room temperature (25 °C), the mixture is peeled from the mold and immersed in DI water at 80 °C for 8 h to dissolve the sugar. Finally, the resulting elastomer foam is dried in an oven at 80 °C for 2 h. This fabrication process ensures an open cell structure with well-defined thickness and smoothness[37,38].

## Mechanical characterization of the silicone-foam

Uniaxial tensile test is carried out on the silicone-foam sample and results are compared with bare silicone in Supplementary Fig. 4a. The elastic modulus of the porous silicone foam is about 7 kPa, whereas the bare silicone layers have a modulus of 1 MPa.

## SF sensor fabrication

To fabricate the foam sensor, a PET film (thickness of 125 $\mu$m) is placed on the vacuum plate (see Supplementary Fig. 13). Then a sacrificial layer of PAA 5% solution is cast on the PET film using a wire bar applicator. A thin film of Sylgard 186 (300 $\mu$m) is cast on the PAA using a film applicator (Zehntner, ZUA2000). The elastomer layer is cured in the oven at 80 °C for 1 h. After activating the surface of the elastomer with oxygen plasma, silver paste (Ag 520 EI from Chimet S.p.A.) is blade cast on the elastomer through a 25 $\mu$m thick Mylar mask and cured in the oven at 80 °C for 1 h. This creates the shielding layer. Subsequently, the insulation layer is cast the shielding layer using the film applicator and cured in the oven at 80 °C for 1 h. Similar to the shielding layer fabrication, the excitation/sense electrodes are cast through the Mylar mask and cured in the oven. This composite layer is then cut and removed from the PET film by dissolving the sacrificial layer in the hot DI water. To bond the foam to the sensor, 300 $\mu$m thick Ecoflex-0030 is cast on top of the electrode layer which has the sense electrode (see Supplementary Fig. 13). After placing the foam on top, the elastomer is cured in the oven at 80 °C for 1 h. Then Ecoflex-0030 is cast on top of the electrode layer, which has the excitation electrode, and the previously cured part is placed on top to bond the other side of the foam. The final step is to cure the elastomer in an oven at 80 °C for 1 h.

## LM sensor fabrication

The fabrication process flow for LM sensors is illustrated in Supplementary Fig. 14. The fabrication starts by molding the microstructured silicone layer (middle layer) in an acrylic master mold. A mixture of Ecoflex-0030 with a weight ratio of 1:1 is prepared and poured into the mold and cured in an oven at 80 °C for 1 h. The bottom layer of the

sensor is composed of a conductive shielding layer and a silicone layer, i.e., silicone + conductive silicone. To fabricate this layer, a conductive ink is prepared by mixing 0.4 g of carbon particles (Ketjenblack EDJ-300) with 11 g of isopropanol (IPA) and 2 g of Ecoflex-0030 part A. This mixture is first ball mixed at 2000 rpm for 2 min. Then 2 g of part B is added and mixed for an additional 2 min. The mixture is cast on a PET substrate using a blade caster with a gap of 250 μm (final film thickness is 54 μm). This conductive silicone mixture is cured in the oven at 80 °C for 1 h. Then a mixture of Ecoflex-0030 with a ratio of 1:1 is prepared separately and casted on the top of the already-cured conductive silicone with a gap of 500 μm. This layer has a final thickness between 276 and 300 μm. The next step is to bond this bottom layer to the molded middle layer using uncured Ecoflex-0030 as the bonding material. A thin layer of Ecoflex is cast on CB+Ecoflex with a gap of 125 μm. After casting, the molded layer is placed on the newly casted Ecoflex and the layers are bonded by curing the silicone in the oven at 80 °C for 1 h. Although the top layer could be fabricated with the same procedure as the bottom layer, we fabricate top layers in two steps. This allows one side of the sensor to be transparent and enables visual inspection during liquid-metal filling. For the top layer, an Ecoflex mixture (1:1 ratio) is casted on a PET substrate, cured in the oven, and bonded to the top side of the middle layer using the same fabrication processes (uncured Ecoflex for bonding). At this step, the microfluidic channels are completely sealed, and one side of the sensor is shielded (black) and the other side is still unshielded (transparent). The next step is filling the microfluidic channels with the liquid-metal, using a vacuum filling method[39]. The top shielding layer is fabricated separately by casting composite layers of carbon-mixed Ecoflex (~54 μm final thickness) and bare Ecoflex (~150 μm final thickness). This layer is then bonded to the sensor using uncured Ecoflex as described above. Finally, the electrical connection to the liquid-metal electrodes and shielding are made using coaxial cables (model 9436 from Alpha wire). The central conductor of the coaxial cable is inserted into the LM channels, and the cable shield is connected to the sensor shield (see Supplementary Fig. 15). We use conductive epoxy to ensure good contact between the cable and sensor shields. An additional grounding cable is used for the shielding layer. All connections are sealed with silicone epoxy and covered with heat shrink to enable a mechanically robust connection.

## Data availability
The datasets generated in this study are available in Zenodo repository at https://doi.org/10.5281/zenodo.6541338.

## Code availability
The custom codes that are used to visualize the data are available in Zenodo repository at https://doi.org/10.5281/zenodo.6541338.

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

## Acknowledgements

This project has received funding from the European Union's Horizon 2020 research and innovation program under grant agreement No 869963. This work was supported by Swiss National Science Foundation under grant number 200020_184661. The authors thank Dr. E. Leroy for his help automating the characterization setup. The authors thank Prof. G. Villanueva, Dr. S.E. Kucuk, and M. Liffredo for their help with the shielding experiments.

## Author contributions

B.A., Y.H., V.C., and H.S. conceived the concepts. V.C., G.G., K.M.D., and H.S. advised on the design and fabrication of the sensors. B.A. and Y.H. conducted experiments and analyzed the data. All authors interpreted the data. B.A. wrote the manuscript. All authors read, edited, and discussed the manuscript and agree with the claims made in this work. H.S. coordinated and supervised the research.

## Competing interests

The authors declare no competing interests.
