## [Peer Review File · Nature Communications]

Shielded Soft Force SensorsREVIEWER COMMENTS

Reviewer #1 (Remarks to the Author):

This paper describes two shielded soft force sensors based on liquid metal channels and silicone foam. The sensors have the ability to sense normal and shear stress and improve accuracy by shielding the capacitors. The mechanism and design tips of the soft shielding layers are discussed. This article has sound scientific motivation and structure and is novel. I suggest publishing this article after a minor revision.

1. Figure 2a lacks the label of a, and the instance color (blue) of the liquid metal in the explosion diagram is not obvious.
2. On page 4, line 110, the authors should explain the reason for the increased resolution and minimum measurement force of the four sensing units.
3. The authors should compare the advantages and disadvantages between liquid metal-based and foam-based sensors.
4. The capacitance variation is about $\sim fF$, how do the authors avoid the parasitic capacitance from the wires?

Reviewer #2 (Remarks to the Author):

In this manuscript, Aksoy and co-workers present an electromechanically shielded soft force sensor. Two different routes have been proposed to develop shielded force sensors: liquid metal-based force sensor made of soft silicone channels filled with a liquid metal and a foam-based force sensor composed of stretchable electrodes and a silicone-elastomer foam. Both types of sensors offer excellent shielding performance and can distinguish normal and shear forces. Various scenarios such as a piece of metal approaching (touching) the sensors and electrostatic adhesive pad attached to the sensors have been used to demonstrate the overall shielding performance of force sensors. A detailed experimentation has also been conducted to show how normal and shear forces can be decoupled. An FEA simulation has also been developed to study the structural deformation as well as shielding behavior of the developed sensor. Finally, the soft sensor has been employed in a soft gripper robot and human hand to monitor and distinguish the normal and shear forces while interacting with external objects.

Overall, the manuscript is well written and presented, and addresses one of the key challenges in highly accurate measurement of forces using soft capacitive sensors, i.e., electromagnetic interference between a capacitive sensor and surrounding objects. However, capacitive, and inductive sensors with electromagnetic shielding performance have been reported by other researchers (Adv. Mater. Technol. 2021, 6, 2100358 and IEEE Robotics and Automation Letters, 2020, 05, 1750, and DOI: 10.1109/ICECS46596.2019.8964922). Various nanoscale materials such as silver nanoparticles, carbon nanotubes, and graphene incorporated into soft polymers have been extensively studied for electromagnetic shielding applications. Additionally, multidimensional or multidirectional force sensing using capacitive sensors have been reported in several publications (DOI: 10.1109/ICRA.2013.6631071, Journal of Microelectromechanical Systems, 2021, 30, 5, 799, and Sensors, 2010, 10, 10211-10225). Therefore, the key contribution and novelty of the present work is not very clear, although the reviewer believes that the shielding performance of the present sensor is better than previously reported ones. The present work requires further systematic and in-depth characterizations in terms of force sensing performance, durability, and shielding effect. Considering these, the reviewer feels that the present manuscript should be reconsidered for publication after major concerns have been addressed by the authors. Some additional comments as below:

- 1- In the foam-based sensor, silver nanoparticle paste has been used for electrodes and shielding. However, cured silver nanoparticle films are prone to cracks and delamination. What is the durability of the sensor when repeatedly stretched or compressed?

- 2- Is there any reason different particles, i.e., carbon and silver particles, have been used for the shielding purpose? Why not other materials like graphene?
- 3- The manufacturing process of both sensors involves multistep and time-consuming fabrication steps. Is the manufacturing process scalable? Can large arrays of sensors be fabricated for electronic skin applications?
- 4- There is no detailed analysis of the shielding performance of the sensor. For example, what is the shielding effectiveness over a wide range of frequencies? How the concentration of carbon particles in silicone is determined for maximum shielding performance?
- 5- In Figure 3a, the capacitance of the unshielded sensor is much noisier compared to the shielded sample during metallic sample approaching them. What is the noise level for both samples when there is no electromagnetic interference?
- 6- Are force sensors sensitive to bending deformations?
- 7- What is the hysteresis performance of the soft force sensor?
- 8- In Figure 3b, the electric field test has obvious effect on the unshielded sensor with negative capacitance change. Why the capacitance change is positive in the case of the shielded sensor?
- 9- Within the manuscript, it is stated that liquid metal-based sensors are made of microchannels. However, Figure 2a clearly shows that the channels are in mm range.
- 10- Label (a) in Figure 2 is missing.

Response to reviews

“Shielded Soft Force Sensors”

by Bekir Aksoy, Yufei Hao, Giulio Grasso, Krishna Manaswi Digumarti, Vito Cacucciolo, and Herbert Shea

We sincerely appreciate the reviewers’ time and comments on our manuscript. We reproduce below the reviewers’ comments in black, provide our responses in green, and indicate changed or new text in red.

Reviewer #1 (Remarks to the Author):

This paper describes two shielded soft force sensors based on liquid metal channels and silicone foam. The sensors have the ability to sense normal and shear stress and improve accuracy by shielding the capacitors. The mechanism and design tips of the soft shielding layers are discussed. This article has sound scientific motivation and structure and is novel. I suggest publishing this article after a minor revision.

We thank the referee for this positive assessment of our work.

1. Figure 2a lacks the label of a, and the instance color (blue) of the liquid metal in the explosion diagram is not obvious.

We thank the referee for pointing out the omission. We added the labeling of this panel and replaced the color palette with a distinct and colorblind-friendly one.

2. On page 4, line 110, the authors should explain the reason for the increased resolution and minimum measurement force of the four sensing units.

In the revised manuscript, we rephrased the text on page 4 (now page 5) to clarify the advantage of having four sensing units for picking up locally applied forces. We removed “increased spatial resolution” statement to avoid any confusion. The new text is replicated below:

“The LM sensors have four of these sensing units which are connected in parallel, i.e., the equivalent capacitance of the sensor is equal to the sum of these sensing units. Because these units cover the entire sensor area, the local forces are picked up by one or multiple units depending on the surface area of the load.”

3. The authors should compare the advantages and disadvantages between liquid metal-based and foam-based sensors.

We added several paragraphs on page 10 to compare/contrast liquid-metal and silicone-foam sensors. We agree this point is needed to help readers decide which path they may wish to follow.

“For a given device area, the LM and SF sensors have comparable performance in terms of force resolution, softness, and resilience under cyclic loading. Both sensor types can withstand high normal forces (> 20 N). However, under high shear forces, the silicone foam sensors fail at approximately 1.5 N, while the liquid-metal sensors can sustain shear loads of over 14 N. The LM structure can more readily be tailored to provide both the required low stiffness and also good robustness.

Fabrication complexity is similar for both sensors, with molding, casting, and bonding steps. The LM sensors require slightly more careful handling during the bonding and the liquid-metal filling steps because the channels must be perfectly sealed. The sensor fabrication process is reproducible: there is less than 2% difference in capacitance between devices from one batch of LM devices. The total thickness of the LM sensor is around 2.2 mm whereas the SF sensors have a thickness of around 1.8 mm.

The LM sensors have better durability than the SF sensors under both normal and shear loads. We therefore used the LM sensors in our demonstrations.”

4. The capacitance variation is about $\sim F$, how do the authors avoid the parasitic capacitance from the wires?

The electrical connections to the sensing electrodes and the shielding were made using coaxial cables (model 9436 from Alpha wire). These shielded cables carry the signal from the sensing electrodes to the AD7746 capacitance-to-digital (CDC) converter chip from Analog Devices, Inc. of the capacitance meter (Capmeter from JLM innovation¹). Separate shielded cables were used for each sensing electrode.

The CDC architecture in the AD7746 measures the capacitance connected between the excitation pin and the input pin (floating capacitive sensing). In theory, any capacitance between these pins to ground should not affect the CDC result. According to the datasheet of AD7746 chip, there is no effect on the measurement (capacitance between the excitation and input pins) if the capacitance between sensing pins to the ground is smaller than 50 pF. The error in the capacitance measurement increases slightly after 50 pF, e.g., 5 fF measurement error occurs at the capacitance of 400 pF.

In our sensors, this parasitic capacitance is less than 30 pF including the capacitance in the wires, so there is in principle no effect of the capacitance due to shielded cables. We added a new figure (Supplementary Fig. 15) that shows the electrical connections and the capacitance meter we used in this study.

¹<https://www.jlm-innovation.de/products/capmeter>

The figure and related text in the manuscript (page 14) are replicated below:

“Finally, the electrical connection to the liquid-metal electrodes and shielding are made using coaxial cables (model 9436 from Alpha wire). The central conductor of the coaxial cable is inserted into the LM channels and the cable shield is connected to the sensor shield (see Supplementary Fig. 15). We use conductive epoxy to ensure good contact between the cable and sensor shields. An additional grounding cable is used for the shielding layer. All connections are sealed with silicone epoxy and covered with heat shrink to enable a mechanically robust connection.”

Supplementary Figure 15. **Electrical connections to the sensors using coaxial cables and the readout using a capacitance meter (Capmeter from JLM innovations GmbH).** Here we show the connection of one coaxial cable to one sensing electrode. Each electrode is connected using a separate cable. The conductor of the coaxial cable is inserted into the LM channel and the cable shield (the ground terminal) is connected to the sensor shield. All connections are sealed with silicone epoxy and covered with heat shrink for mechanical robustness. The shielded cables carry the signal from the sensing electrodes to the AD7746 capacitance-to-digital (CDC) converter chip (from

Analog Devices, Inc.) of the capacitance meter (from JLM innovation¹). The CDC architecture used in the AD7746 measures the capacitance connected between the excitation pin and the input pin (floating capacitive sensing). In theory, any capacitance of less than 50 pF from these pins to ground should not affect the CDC output.

Reviewer #2 (Remarks to the Author):

In this manuscript, Aksoy and co-workers present an electromechanically shielded soft force sensor. Two different routes have been proposed to develop shielded force sensors: liquid metal-based force sensor made of soft silicone channels filled with a liquid metal and a foam-based force sensor composed of stretchable electrodes and a silicone-elastomer foam. Both types of sensors offer excellent shielding performance and can distinguish normal and shear forces. Various scenarios such as a piece of metal approaching (touching) the sensors and electrostatic adhesive pad attached to the sensors have been used to demonstrate the overall shielding performance of force sensors. A detailed experimentation has also been conducted to show how normal and shear forces can be decoupled. An FEA simulation has also been developed to study the structural deformation as well as shielding behavior of the developed sensor. Finally, the soft sensor has been employed in a soft gripper robot and human hand to monitor and distinguish the normal and shear forces while interacting with external objects.

Overall, the manuscript is well written and presented, and addresses one of the key challenges in highly accurate measurement of forces using soft capacitive sensors, i.e., electromagnetic interference between a capacitive sensor and surrounding objects. However, capacitive, and inductive sensors with electromagnetic shielding performance have been reported by other researchers (Adv. Mater. Technol. 2021, 6, 2100358 and IEEE Robotics and Automation Letters, 2020, 05, 1750, and DOI: 10.1109/ICECS46596.2019.8964922). Various nanoscale materials such as silver nanoparticles, carbon nanotubes, and graphene incorporated into soft polymers have been extensively studied for electromagnetic shielding applications. Additionally, multidimensional or multidirectional force sensing using capacitive sensors have been reported in several publications (DOI: 10.1109/ICRA.2013.6631071, Journal of Microelectromechanical Systems, 2021, 30, 5, 799, and Sensors, 2010, 10, 10211-10225). Therefore, the key contribution and novelty of the present work is not very clear, although the reviewer believes that the shielding performance of the present sensor is better than previously reported ones. The present work requires further systematic and in-depth characterizations in terms of force sensing performance, durability, and shielding effect. Considering these, the reviewer feels that the present manuscript should be reconsidered for publication after major concerns have been addressed by the authors. Some additional comments as below:

We thank the reviewer for the overall positive assessment of our work, and address below in detail the points that were raised.

We carried out additional experiments to provide a more in-depth characterization of the sensors. We first tested the sensor performance under cyclic load (0 to 2 N for over 10^4 cycles). We then measured the mechanical hysteresis of the sensors and the capacitance change during loading and unloading at different speeds. We developed a test setup to continuously bend the sensor, and measured the sensors' sensitivity vs. radius of curvature.

In addition to our previous shielding experiments, we now include new shielding effectiveness measurements of only the shielding layer, using a network analyzer over a broad frequency range. New figures containing the experimental data of these experiments are included in the revised manuscript. These figures are also provided below along with the related comments.

Regarding novelty: capacitive-based multi-axis force sensors have indeed been reported in the literature. However, these multi-axis sensors lack shielding against electromagnetic interference, making them often impractical when conducting objects (even a hand or arm) are moved in proximity. A key advantage of our soft sensor is that it can be reliably used in a very wide range of soft robotic applications, sensing normal and shear forces.

There have been some recent publications reporting shielding of flexible sensors based on a printed circuit board (PCB) enclosed within a soft conductive layer²⁷⁻²⁹. Those designs use an elastomer-based conductive layer to serve as both the shield and also one of the sensing electrodes. This configuration gives shielded flexible sensors but is limited to measuring normal forces only. Even with additional electrodes, a design where the continuous shield also serves as one of the sensing electrodes cannot be adapted for shear force measurement.

A solution to this is to decouple the shielding electrode from the sensing units, which is the approach we followed in this paper. However, the decoupling introduces a new challenge in the form of parasitic capacitance between the shield and sensing electrodes, problematic because this capacitance changes when a force is applied to the sensor (the sensors are completely soft). A key contribution of our work is two designs to overcome the effect of this parasitic capacitance and reliably measure both normal and shear forces in shielded soft sensors.

In the revised manuscript we added new explanatory text on page 2 to more clearly distinguish our work from the aforementioned studies. This text is replicated below in red.

“Several recent publications report shielded capacitive sensors based on a flexible printed circuit board (PCB) enclosed within a compliant conductive layer²⁷⁻²⁹. The external soft conductive layer to serve as both the shield and as one of the sensing electrodes. This configuration is intrinsically limited to measuring normal force. Measuring shear forces, requires sensing the relative change in capacitance of multiple (at least two) capacitors³⁰⁻³². Even with additional electrodes, a design where the continuous shield also serves as one of the sensing electrodes cannot be adapted for shear force measurement. A solution to this is to decouple the shielding electrode from the sensing units, which is the approach we follow in this article. This introduces a new challenge in the form of parasitic capacitance between the shield and sensing electrodes, which is problematic because this capacitance changes when a force is applied to the sensor, as the sensors are completely soft. A key contribution of our work is two designs to overcome the effect of this parasitic capacitance and reliably measure both normal and shear forces in shielded soft sensors.”

1- In the foam-based sensor, silver nanoparticle paste has been used for electrodes and shielding. However, cured silver nanoparticle films are prone to cracks and delamination. What is the durability of the sensor when repeatedly stretched or compressed?

Silver nanoparticles can crack when they are coated on silicone films, which could lead to delamination between the layers. To avoid this, we first treated the surface of the silicone film with O₂ plasma, then printed the silver-based electrodes. Additionally, we cast another silicone layer on the top of the silver electrodes to prevent delamination under high stretching. This way, the silver nanoparticles are firmly embedded within silicone films.

In the revised manuscript, we provided new experimental results (in Supplementary Fig. 11) for the performance of both SF and LM sensors under cyclic deformations. The capacitance of the sensors was measured over thousands of cycles over 30 h for an applied force of 2 N on the LM sensor and of 0.8 N on the SF sensor. The maximum deviation in the capacitance was <10 fF in the LM sensor and <12 fF in the SF sensor. These variations correspond to 35 mN in the LM sensor and 53 mN in the SF sensor. We observed that capacitance fluctuations correlate with the temperature of the capacitance-to-digital converter chip of the capacitance meter, and is thus only in part due to the potential aging of the devices.

The figure and related text in the revised manuscript (on page 10) are replicated below:

“The sensors are resilient to cyclic loading. The capacitance of the sensors was measured over thousands of cycles over 30 h for an applied force of 2 N on the LM sensor and of 0.8 N on the SF sensor (see Supplementary Fig. 11). The maximum deviation in the capacitance from initial value after 10⁴ cycles was <10 fF in the LM sensor and <12 fF in the SF sensor. These variations correspond to 35 mN in the LM sensor and 53 mN in the SF sensor.”

Supplementary Figure 11. Performance of the sensors under cyclic loading. **a** Capacitance change of the LM sensor under an applied force of 2 N for the first five cycles. **b** Capacitance change of the LM sensor under an applied force of 2 N and at 0 N vs. cycle number for 10^4 cycles. The capacitance difference between the first and last cycles is less than 10 fF, corresponding to <35 mN force. **c** The capacitance of the SF sensor under an applied force of 0.8 N vs. time for the first nine cycles. **d** Capacitance change of the SF sensor under an applied force of 0.8 N and at 0 N vs. cycle number for 10^3 cycles. The capacitance difference between the first and last cycles is less than 12 fF, corresponding to <53 mN force.

2- Is there any reason different particles, i.e., carbon and silver particles, have been used for the shielding purpose? Why not other materials like graphene?

We chose the shielding materials for sufficient conductivity, our technical know-how and experience, and ink availability, and importantly based on their compatibility with the other sensor materials, i.e., good adhesion to the other layers, and an elasticity modulus of ~ 1 MPa that matches the other layers.

The silver ink is very conductive (0.73×10^6 S) and is commercially available. It comes in a paste form and can be directly coated on the silicone surface. The Carbon-based ink has been developed in our laboratory for more than a decade for various projects (mostly for DEAs). The carbon-PDMS ink has a very strong bonding to silicones and has a comparable conductivity (0.37×10^6 S) to silver ink. Since we use the same silicone material (PDMS) for the sensor and for the shield, we don't observe any cracking or delamination, even at very high stretching. The materials are both proven to effectively block the proximity and E-field interference.

Graphene could be used as a conductor, but we would have to develop a process of mixing and dispersing graphene with PDMS, and ensure strong adhesion of this conductive composite to the sensor, as the shielding layer needs to be robust against abrasion and mechanical wear.

3- The manufacturing process of both sensors involves multistep and time-consuming fabrication steps. Is the manufacturing process scalable? Can large arrays of sensors be fabricated for electronic skin applications?

Though the fabrication process involves multiple steps, these steps are simple. For instance, patterning the LM sensors is done by molding (for the central section), blade casting (for top and bottom layers), and by bonding. One could scale up the number of the sensors by increasing the number of microfluidic channels, which can be achieved simply by redesigning the master mold. The master mold can be fabricated using a range of widely available techniques such as 3D printing, laser cutting, or photolithography (if higher resolution is desired). Designing the master mold allows arraying and also different sensor size.

Additionally, the number of sensors is independent of the fabrication steps, i.e., the number of fabrication steps is the same for a single unit and for an array. Working up to A4 sizes are possible with our university equipment. Scaling up to larger area is possible with industrial-sized tooling.

4- There is no detailed analysis of the shielding performance of the sensor. For example, what is the shielding effectiveness over a wide range of frequencies? How the concentration of carbon particles in silicone is determined for maximum shielding performance?

Our objective was to show the effectiveness of the shielding integrated with the sensor for several sensing use-cases, including showing immunity to the motion of conductors near the soft sensors, and correct operation even when only a few 100 μm away from square-wave switching kV signals. In practice, the soft shielding is sufficient for even extreme use cases.

Following the referee's suggestion, we tested the shielding properties of the shielding membranes (PDMS with either silver or carbon conducting layer) using a network analyzer (NA), similar to how Yoo *et. al.*²⁷ measured their shielding layers. We placed different the membrane between the network analyzer probes and compared the signal attenuation (i.e., performed a S_{21} measurement). We present the experimental data in a new figure (Supplementary Fig. S7).

The concentration of the carbon particles is based on a compromise between EM shielding and adhesion of the conductive composite to the bare silicone layers. Low carbon content leads to better bonding but to lower conductivity. Higher carbon concentration results in higher conductivity (i.e., better shielding) but in poorer adhesion and lower stretchability. We can compensate to some extent for the conductivity of the carbon-loaded layer by increasing its thickness, at the expense of slightly higher stiffness.

The figure and related text in the revised manuscript (on page 7) are replicated below:

“The shielding effectiveness of the carbon-loaded silicone, silver-based ink, and bare silicone layers was measured using a network analyzer (E5071C from Agilent Technologies). Two ports of the network analyzer were fixed 0.5 mm apart, as shown in Supplementary Fig. 7a. The membranes were placed between these probes, and the S_{21} (i.e., transmission) characteristics were measured as the frequency was swept from 10 MHz to 200 MHz. A baseline test was done without any membrane between the probes. Then the bare, carbon-based, and silver-based silicone layers were used. The silver-based shielding performs the best by attenuating the signal by 19 dB to 30 dB, whereas carbon-based shielding reduces the signal by 18 dB to 20 dB.”

Supplementary Figure 7. Measurement of transmission of different shielding layers using a network analyzer (E5071C from Agilent Technologies). **a** Photograph and schematic describing the experimental setup to measure the shielding properties of circular thin films. Two probes of the network analyzer are clamped with a spacing of approximately 0.5 mm. The film to be tested is placed between them. The S_{21} parameter is measured while the frequency is swept from 10 MHz to 200 MHz. **b** Signal attenuation for different materials plotted vs frequency. The case of: no membrane between the probes, bare silicone membrane, carbon-based silicone composite membrane, and silver-based silicone membrane are compared. The silver-based shielding performs the best, attenuating the signal by 19 dB to 30 dB compared to bare PDMS, whereas the carbon-based shielding reduces by 18 dB to 20 dB compared to bare PDMS.

5- In Figure 3a, the capacitance of the unshielded sensor is much noisier compared to the shielded sample during metallic sample approaching them. What is the noise level for both samples when there is no electromagnetic interference?

Thanks to the grounded shielding layers the noise (standard deviation of the capacitance change over 20 s of measurements) in the shielded sensors is much lower than in the unshielded sensors. When the sensors are at the rest position, the variation in capacitance of the shielded sensor is ± 0.09 fF whereas the unshielded sensor has ± 2.6 fF variation in the capacitance. This noise difference between the sensors is obvious in Figure 3b at 0 kV electric field.

6- Are force sensors sensitive to bending deformations?

We thank the referee for this question. This is an important point: since our sensor is soft, it is designed to be applied to non-flat surfaces. So we now report the sensor's sensitivity to bending. For this test, we placed the sensor on a bendable beam where we continuously changed and tracked the bending radius of curvature of the sensor. A new figure (Supplementary Fig. 12) is added to plot the evolution of the capacitance as a function of the curvature of the sensor.

The test was repeated for two configurations: with the bending axis orthogonal to the shear direction and with the bending axis parallel to the shear direction. Supplementary Fig. 12d shows the evolution of the capacitances as we change the curvature of the sensor. The measured change of capacitance per curvature was 2.67 pF/mm^{-1} and 1.64 pF/mm^{-1} for the orthogonal and parallel configurations, respectively. For example, if a normal load of 1N is applied and the sensor bends with a change in curvature of 0.01 mm^{-1} , (this corresponds to 100 mm change in the radius of curvature, from 160 mm to 60 mm) the capacitance change would be -22 fF and -12 fF, for orthogonal and parallel configurations, respectively. This would cause a force error of approximately 85 mN and 46 mN, small but not negligible compared to the 1N load.

If the sensor is placed on a surface of fixed curvature, the sensor can simply be calibrated for this curved surface. If we however use the sensor in a scenario where the sensor is being simultaneously bent and pressed at the same time,

the capacitances change due to both bending and pressing. Several techniques can be employed to compensate for this to obtain the normal and shear forces. For instance, an additional sensor can be placed on the back of the structure to measure the curvature only, which allows to compensate for the effect of the sensor bending. This way both curvature and applied force can be measured independently. Another approach would be to make the sensor thinner. This way, the deformation in the sensor due to bending motion would be limited and thus show smaller changes in the capacitance.

The figure and related text in the manuscript (on page 10) are replicated below:

“The sensitivity of the sensors to bending was measured using an experimental setup where the bending radius of curvature of the sensor could be continuously varied. The experimental setup and plot of capacitance as a function of the bending curvature are shown in Supplementary Fig. 12. The test was repeated for two configurations: the bending axis is orthogonal to the shear direction and the bending axis is parallel to the shear direction. The measured change of capacitances per curvature change was 2.73 pF/mm^{-1} and 1.73 pF/mm^{-1} for orthogonal and parallel configurations, respectively. For a 100 mm change in the radius of curvature, the bending causes a force error of approximately 85 mN in orthogonal configuration and 46 mN in parallel configuration.

If the sensor is placed on a surface of fixed curvature, the sensor can simply be calibrated for this given curvature, giving the same accuracy in normal and shear force components as for the flat state. If we however use the sensor in a scenario where the sensor is being simultaneously bent and pressed at the same time, the capacitance changes due to bending deformation then need to be taken into account for accurate measurements. Different approaches can be implemented to compensate this effect, such as attaching an additional sensor on the back of the beam that can sense the curvature only. This way, both the applied load and the curvature can be measured simultaneously.”

Supplementary Figure 12. **Sensor sensitivity vs. bending deformation.** **a** Schematic of experimental setup for controlled bending of the sensor. **b** The photographs of the undeformed and bent configurations ($\kappa = 0.03 \text{ mm}^{-1}$) of the sensor. **c-d** The sensor was attached to a bendable beam in two different configurations: the bending axis is orthogonal to the shear direction (blue and red curves) and the bending axis is parallel to the shear direction (yellow

and purple curves). The change of the capacitances as a function of the bending curvature for two different orientations.

7- What is the hysteresis performance of the soft force sensor?

Following the referee's question, we carried out hysteresis analysis on our sensors. We measured the force and capacitance change as we loaded and unloaded the sensors. We repeated the experiments at two different speeds to investigate the effect of the loading rate. The sensor shows a small degree of elastic hysteresis of 5% between the loading and unloading due to viscoelasticity of the PDMS. We use a PDMS with low viscoelastic effects, so this is not very surprising. The hysteresis behavior was observed to be identical for both tested speeds. In the case of capacitance change as a function of displacement, a 2% hysteresis was observed. Once again, there was no rate dependence of the experiment. In all cases, the force and capacitance return to 0 values when the displacement is recovered, showing no permanent deformation.

The sensor thus shows low hysteresis and no permanent set, demonstrating the robustness of the measurements. The small hysteresis does not depend on the loading speed for the tested range. We included a new figure (Supplementary Fig. 10) summarizing these results in the revised supplementary document.

The figure and related text in the revised manuscript (page 9) are replicated below:

"The elastic hysteresis of the sensors was measured using a uniaxial test machine (see Supplementary Fig. 10a). The sensor response was tested at two different speeds: 0.002mm/s and 0.02 mm/s. The sensor shows a small degree of elastic hysteresis of 5% (the maximum of the difference in the ordinate expressed as a percentage over the range of the ordinate) between the loading and unloading due to viscoelasticity of the PDMS (see Supplementary Fig. 10b). The hysteresis behavior was observed to be identical for both tested speeds. In the case of capacitance change as a function of displacement, a 2% hysteresis was observed (see Supplementary Fig. 10c). Once again, there was no rate dependence. In all cases, the force and capacitance return to initial values when the displacement is recovered, showing no permanent deformation. The sensor shows low hysteresis and no permanent set, important features for soft sensors."

Supplementary Figure 10. Hysteresis of force and of capacitance during loading and unloading. **a** Photograph and schematic of the experimental setup used for hysteresis analysis. **b** The graph plots force vs. displacement of the LM sensor taken at two different speeds: 0.002 mm/s and 0.02 mm/s. The sensor has a small degree of elastic hysteresis of 5% and it is the same for the tested loading speeds. **c** The capacitance change during loading and unloading plotted against the applied displacement, showing 2% of hysteresis. The capacitance hysteresis is the same for both loading rates. **d** The capacitance change plotted as a function of the applied load. The capacitance change shows a small difference between loading and unloading curves and is the same for different loading speeds.

8- In Figure 3b, the electric field test has obvious effect on the unshielded sensor with negative capacitance change. Why the capacitance change is positive in the case of the shielded sensor?

The capacitance of the shielded sensor at the rest state (no force is applied) changes from 0.03 ± 0.08 fF (mean value ± 1 standard deviation over 20 s of measurements) to 0.54 ± 0.10 fF for the applied voltages of 0 kV and for 1.5 kV, respectively. At the same voltages, the capacitance of the unshielded sensor changes from -5.38 ± 2.58 fF and -34.88 ± 23.39 fF. The capacitance variation in the shielded sensors is very small (corresponding to 1.35 ± 0.25 mN of normal force and 0.11 ± 0.02 mN of shear force) and may not be directly related to the electric field generated by the electroadhesive patch but might be due to the electronics used to readout for the capacitance, i.e., the temperature variation of the capacitance-to-digital chip.

9- Within the manuscript, it is stated that liquid metal-based sensors are made of microchannels. However, Figure 2a clearly shows that the channels are in mm range.

The liquid metal sensors have two different channels: the channels in the top row have a cross-section of $500 \mu\text{m} \times 500 \mu\text{m}$ and the bottom channels have a cross-section of $500 \mu\text{m} \times 1.4 \text{ mm}$. The design parameters of the sensors are provided in Supplementary Fig. 3. Although some dimensions are in the range of mm, most design parameters are in micrometer range. Therefore, we refer them as microfluidic channels.

10- Label (a) in Figure 2 is missing.

We have corrected this omission.

References

27. Yoo, D., Won, D. J., Cho, W., Lim, J. & Kim, J. Double Side Electromagnetic Interference-Shielded Bending-Insensitive Capacitive-Type Flexible Touch Sensor with Linear Response over a Wide Detection Range. *Advanced Materials Technologies* **6**, (2021).
28. Wang, H., Totaro, M. & Beccai, L. Development of fully shielded soft inductive tactile sensors. in *2019 26th IEEE International Conference on Electronics, Circuits and Systems (ICECS)* 246–249 (2019).
29. Won, D. J., Yoo, D. & Kim, J. Effect of a Microstructured Dielectric Layer on a Bending-Insensitive Capacitive-Type Touch Sensor with Shielding. *ACS Applied Electronic Materials* **2**, 846–854 (2020).
30. Fernandes, J., Chen, J. & Jiang, H. Three-Axis Capacitive Sensor Arrays for Local and Global Shear Force Detection. *Journal of Microelectromechanical Systems* **30**, 799–813 (2021).
31. Roberts, P., Damian, D. D., Shan, W., Lu, T. & Majidi, C. Soft-matter capacitive sensor for measuring shear and pressure deformation. in *Proceedings - IEEE International Conference on Robotics and Automation* 3529–3534 (2013). doi:10.1109/ICRA.2013.6631071.
32. Cheng, M. Y., Lin, C. L., Lai, Y. T. & Yang, Y. J. A polymer-based capacitive sensing array for normal and shear force measurement. *Sensors (Switzerland)* **10**, 10211–10225 (2010).

REVIEWERS' COMMENTS

Reviewer #1 (Remarks to the Author):

The authors have addressed the comments. I recommend publishing this work. Thank you.

Reviewer #2 (Remarks to the Author):

The authors have made significant improvements in their revised manuscript including experiments related to shielding performance, bending experiment, and durability test. After a careful consideration, the review believes the current version of the manuscript is acceptable for publication.